# Zn^2+^ and Cu^2+^ Interaction with the Recognition Interface of ACE2 for SARS-CoV-2 Spike Protein

**DOI:** 10.3390/ijms24119202

**Published:** 2023-05-24

**Authors:** Alessio Pelucelli, Massimiliano Peana, Bartosz Orzeł, Karolina Piasta, Elzbieta Gumienna-Kontecka, Serenella Medici, Maria Antonietta Zoroddu

**Affiliations:** 1Department of Chemical, Physical, Mathematical and Natural Sciences, University of Sassari, 07100 Sassari, Italy; alessiopelucelli@gmail.com (A.P.); sere@uniss.it (S.M.); zoroddu@uniss.it (M.A.Z.); 2Faculty of Chemistry, University of Wroclaw, 50-383 Wroclaw, Poland; bartosz.orzel@chem.uni.wroc.pl (B.O.); elzbieta.gumienna-kontecka@chem.uni.wroc.pl (E.G.-K.)

**Keywords:** ACE2, peptides, zinc complexes, copper complexes, metal interaction, potentiometry, spectroscopy, NMR

## Abstract

The spike protein (S) of SARS-CoV-2 is able to bind to the human angiotensin-converting enzyme 2 (ACE2) receptor with a much higher affinity compared to other coronaviruses. The binding interface between the ACE2 receptor and the spike protein plays a critical role in the entry mechanism of the SARS-CoV-2 virus. There are specific amino acids involved in the interaction between the S protein and the ACE2 receptor. This specificity is critical for the virus to establish a systemic infection and cause COVID-19 disease. In the ACE2 receptor, the largest number of amino acids playing a crucial role in the mechanism of interaction and recognition with the S protein is located in the C-terminal part, which represents the main binding region between ACE2 and S. This fragment is abundant in coordination residues such as aspartates, glutamates, and histidine that could be targeted by metal ions. Zn^2+^ ions bind to the ACE2 receptor in its catalytic site and modulate its activity, but it could also contribute to the structural stability of the entire protein. The ability of the human ACE2 receptor to coordinate metal ions, such as Zn^2+^, in the same region where it binds to the S protein could have a crucial impact on the mechanism of recognition and interaction of ACE2–S, with consequences on their binding affinity that deserve to be investigated. To test this possibility, this study aims to characterize the coordination ability of Zn^2+^, and also Cu^2+^ for comparison, with selected peptide models of the ACE2 binding interface using spectroscopic and potentiometric techniques.

## 1. Introduction

The human angiotensin-converting enzyme 2 (ACE2) receptor plays a critical role in regulating blood pressure, but it has gained notoriety as the key entry point for the SARS-CoV-2 virus that causes COVID-19. This protein acts as a receptor for the virus, allowing it to enter and infect human cells [1]. The ACE2 receptor is expressed in various tissues including the lungs, heart, and kidneys, making it a crucial target for the virus to establish a systemic infection [2].

ACE2 receptor is a type I transmembrane protein belonging to the family of carboxypeptidase enzymes. It carries out several functions, including regulating blood pressure by converting angiotensin II into angiotensin-(1–7), which has vasodilatory effects. Additionally, it exerts a protective effect on the heart and kidneys, making it a potential therapeutic target to treat cardiovascular and renal diseases [3].

The SARS-CoV-2 spike protein (S) interacts with the ACE2 receptor acting as a key and allowing the virus to enter and infect human cells. The spike has a trimeric structure composed of three identical subunits and contains two domains, the N-terminal S1 domain, and the C-terminal S2 domain. The S1 domain mediates S protein binding to the ACE2 receptor, while the S2 domain is responsible for the fusion between virus and host cell membranes [4]. The interaction between the spike protein and the ACE2 receptor is now well documented. The Spike protein binds to the ACE2 receptor with high affinity, allowing the virus to enter host cells and replicate therein [1,2,5]. Such interaction results in the loss of ACE2 function and an increase in angiotensin II levels, leading to a state of hypertension and increased oxidative stress, which contributes to the severity of COVID-19. Studies have shown that the number of ACE2 receptors in certain tissues can affect the severity of COVID-19 [6]. For example, individuals with higher levels of ACE2 expression in the lungs are more susceptible to severe respiratory distress, while higher expression in the heart can lead to a higher risk of cardiovascular complications. Moreover, certain genetic variations in the ACE2 gene have been associated with an increased risk of severe disease in COVID-19 patients [6].

The ACE2-S bond is a critical aspect of the ability of the SARS-CoV-2 virus to infect human cells and cause COVID-19 [7], since their interaction is specific and highly affinity driven [8]—much higher than other coronaviruses [9]. This specificity is critical for the virus to establish a systemic infection and cause disease.

The structure of the ACE2 receptor is also critical for the interaction with the S protein. ACE2 receptor exists in multiple conformations, and the preferred conformation for S binding has been identified as a closed and compact form. This conformation allows for optimal interaction between the amino acids of the ACE2 receptor and the S1 domain and provides the highest binding affinity between the two proteins. The exact details of the preferred conformation of the ACE2 receptor, such as the specific arrangement of amino acids and its interactions, are still the subject of ongoing research. Mutations in the S protein can affect its ability to bind the ACE2 receptor and reduce the infectivity of the virus [10], but also the other way around.

Understanding the details of this interaction is crucial for the development of therapies to prevent and treat COVID-19. In fact, although the massive vaccination campaigns significantly decreased the mortality associated with COVID-19 all over the world and the virus is now circulating freely with a severity comparable to a normal cold or flu, there is still a part of the population suffering from severe complications after infection, and the emergence of new mutations cannot be excluded [11]. Thus, further research is needed to fully understand the mechanisms behind the spike-ACE2 receptor interaction and how it contributes to the severity of the disease.

The specific amino acids involved in the interaction between the S protein and the ACE2 receptor have been identified. ACE2-S binding is primarily driven by the S1 domain of the S protein, and several amino acids at this interface seem to play a critical role in the recognition and binding process [12].

An analysis of the structural basis of receptor recognition by SARS-CoV-2 evidenced that residues N487, Q493, Y505, Y499, T500 and G446 of RBD (receptor-binding domain) in the S protein are the recurrent amino acids which can make a direct contribution to the affinity of the ACE2-S interaction (Figure 1) [12,13]. These residues are involved in hydrogen bonding with the ACE2 receptor, and mutations in these positions seem to reduce the binding affinity between the two proteins [14].

Likewise, also some specific residues in the ACE2 receptor play a crucial role in the interaction: Q24, D30, H34, E35, E37, D38, Y41, and Q42 (Figure 1) [15,16]. These amino acids are involved in the formation of a network of hydrogen bonds with the specific residues of the S protein, already mentioned, and mutations at these positions also seem to weaken the binding between the two proteins. These residues are located in the C-terminal part of ACE2 which represents the main interface region between ACE2 and S: S_19_T_20_I_21_E_22_E_23_Q_24_A_25_K_26_T_27_F_28_L_29_D_30_K_31_F_32_N_33_H_34_E_35_A_36_E_37_D_38_L_39_F_40_Y_41_Q_42_.

A growing body of evidence supports that zinc deficiency is one of the major determinants of COVID-19 risk and severity [17]. Zn^2+^ binds to specific residues (H_374_E_402_XXH_378_) in ACE2 modulating its activity [18], and is also able to modify the conformation of the receptor [19]. What is particularly interesting in the interface region of ACE2 is the abundance of amino acids such as aspartates, glutamates, and histidine with peculiar and selective coordinating abilities. With regard to the interaction between ACE2 and the S protein of SARS-CoV-2, it is thought that the coordination of metal ions could alter the conformation of ACE2 and thus modify its affinity for the S protein. In particular, the ability of the human ACE2 receptor to coordinate metal ions in the same region where it binds to the S protein of SARS-CoV-2 could have a crucial impact in the mechanism of recognition and interaction of the two proteins, with repercussions on their binding affinity that deserve to be investigated. We preliminarily tested the ability of the interface region of ACE2 to bind Zn^2+^ ions using MIB2, a metal ion binding site prediction and modeling server, which performs metal ion docking and predicts metal binding residues from deposited X-ray structures [20]. The results were promising, as we could observe that a recurrent interaction pattern was present even considering different ACE2 and ACE2-S structures [21,22,23,24,25]. In particular, the region with a high Zn^2+^ binding probability involved amino acids comprising the 30–37 sequence, which consists of a rich set of coordinating residues, D_30_KFNHEAE_37_ (Figure 2).

In order to test this hypothesis, the present study aims to characterize the coordination ability of Zn^2+^ ions with selected peptide fragments of the binding interface of ACE2. In particular, we selected three sequence models: P29-38 (Ac-L_29_DKFNHEAED_38_-NH_2_), P23-42 (Ac-E_23_QAKTFL_29_DKFNHEAED_38_LFYQ_42_-NH_2_), and P19-42 (Ac-S_19_TIEEQAKTFL_29_DKFNHEAED_38_LFYQ_42_-NH_2_). Peptide P29-38 contains, according to MIB2 prediction, the interacting sequence able to bind Zn^2+^. The other two peptides, P23-42 and P19-42, are larger models of the recognition region of the two proteins, and both include the sequence of P29-38. The longest peptide, P19-42, represents the whole binding interface of ACE2. We studied the binding affinity of the three peptides models for Zn^2+^, through a combination of potentiometric and spectroscopic techniques. Moreover, we include in the study also Cu^2+^, for comparison purposes, and to gain more insight into the metal coordination ability of this crucial ACE2 domain. Molecular modeling techniques were further used to elaborate three-dimensional models of the main Zn(II) and Cu(II) complexes on the basis of the experimental evidences.

## 2. Results and Discussion

### 2.1. Protonation Equilibria

The deca-peptide Ac-L_29_DKFNHEAED_38_-NH_2_ (P29-38) can be considered a H_6_L ligand with six protonation constants in the pH range investigated (Table 1 and Figure 3). 

The first two pKa values (3.15 and 3.83) correspond to the deprotonation of the carboxylic group of Asp30 and Asp38 residues. The next two pKa values (4.38 and 4.93) arise from the deprotonation of Glu35 and Glu37. Potentiometric data cannot distinguish between the two Asp and Glu residues. The following two pKas (6.84 and 10.21) are the result of the deprotonation of His34 and Lys31 sidechains, in good agreement with the literature values of similar systems [26,27]. NMR spectra recorded for free peptide P29-38 for a wide range of values corroborated these deprotonation steps. Chemical shift changes following the increase in pH were detected for Asp and Glu nuclei at pH below 5, whereas for the His residue, large chemical shift changes were detected from pH 6 to 7.4, and for Lys at pH > 10 (Appendix A). Unfortunately, both longer peptides, P23-42 and P19-42, were very poorly soluble in water; therefore, it was not possible to perform a potentiometric analysis. Consequently, their coordination ability towards Zn^2+^ and Cu^2+^ was evaluated through NMR spectroscopy only.

### 2.2. Zinc(II) Complexes

Peptide P29-38 forms four species with Zn(II) ions (Table 2 and Figure 4). 

The first species is [ZnH_2_L], which starts from pH 3.5 and reaches only 28% of the maximum Zn^2+^ concentration in solution at pH 5.5. At this pH, all the carboxylic groups are almost completely deprotonated, while His34 and Lys31 residues are still protonated. The proposed coordination mode of Zn^2+^ could involve both aspartic residues, Asp30 and Asp38, and one or both glutamate residues (Glu35 and/or Glu37). This species is soon replaced, starting from pH 4.5, by [ZnHL]^−^, which is present in solution up to pH 9.5, with a maximum at pH 7 (about 75% of Zn^2+^ in solution). Zn^2+^ binding should involve, for this species, His34, Asp30, Asp38, and Glu37 residues. This coordination mode, through His34 bound to the metal ion, prevents, in fact, Glu35 coordination, as its sidechain protrudes towards the opposite side of the complex. The last two species are [ZnLH_-1_]^3−^, that starts forming from pH 6.5 to pH 11, with a maximum formation at pH 9, and [ZnLH_-2_]^4−^, that starts forming from pH 8 with a maximum at pH 11. The deprotonation could involve, for the first species [ZnLH_-1_]^3−^, two water molecules, and for [ZnLH_-2_]^4−^, it could involve the ammonium group of Lys31.

Nuclear magnetic resonance spectroscopy was used to investigate the behavior of Zn^2+^ towards P29-38 in a wide range of pHs as a supporting tool to the potentiometric results. The complete chemical shift assignment of proton and carbon nuclei at several pH values for the free peptide P29-38 was performed with the help of complementary NMR techniques 2D ^1^H-^1^H TOCSY, ^1^H-^13^C- HSQC, and ^1^H-^1^H ROESY. The Zn(II)-P29-38 system was analyzed at pH 5, 5.5, 6, 7.4, 8, and 9. Figure 5 and Figure 6 report a comparison of the free peptide and the Zn(II)-P29-38 system for selected regions of TOCSY and HSQC spectra at pH 5 and 6, respectively. The red labeled signals refer to those perturbed after interaction with the metal ion.

Since the first species [ZnH_2_L] is present at a very low concentration, the spectra at pH 5 showed very little changes after Zn^2+^ addition. By increasing the pH up to 6, different perturbations can be observed both in the TOCSY and HSQC spectra. These perturbations are almost all related to the coordinating residues Asp30, Asp38, and Glu37, in agreement with the potentiometric previsions. The binding of Zn^2+^ to these residues also leads to selected perturbations on other nearby residues, such as Lys31, Phe32, and Ala36, which are indirectly affected by complex formation since they experience a new electronic environment due to the subsequent conformational change of the peptide. Furthermore, His34 is also affected when the pH is increased, in agreement with its involvement in metal binding with the following species [ZnHL]^−^, that starts appearing from pH 4.5, while at pH 6 it reaches the same concentration in solution as the first species. An analysis of the NMR spectra recorded at pH 7.4 gave us the full information regarding the following species. Figure 7 reports some significant examples of selective perturbations in the ^1^H-^13^C HSQC spectra of the residues involved in the binding with Zn^2+^ (Figure 7a). Furthermore, in Figure 7b, the histogram relating to the variations in the chemical shifts between the Zn(II)-P29-38 system and the free one for the H and C nuclei is reported. The scheme in Figure 7c gives a visual indication of the major perturbations that have been highlighted through the analysis of the NMR spectra at pH 7.4, and Figure 7d shows a proposed structural model of this system.

Specific residues are perturbed in accordance with the potentiometric evaluation, suggesting a {(Asp), Asp, Glu, His} coordination mode. In particular, the greatest variations in the chemical shift values are related to the nuclei in the sidechain of His34, Glu37, Asp30, and Asp38. The NMR data confirm the Zn^2+^ binding to the imidazole ring of histidine. The trends in Δδ for Glu37 (QG>HB2/3>HA) and for Asp30 and Asp38 (HB2/3>HA) indicate the involvement of metal binding through their carboxylic groups. Glu35 is not involved in the binding. Structural models were built for this species and are depicted in Figure 7d. The Zn(II): P29-38 system at pH 8 and 9 resembles that at pH 7.4, in agreement with the conclusion that the species [ZnH_-1_L]^−3^ maintains the same coordination sphere of [ZnLH]^−^ (Appendix A). 

Despite the low solubility, some useful information about the coordination ability of the P23-42 peptide towards Zn^2+^ has been evaluated through NMR spectroscopy at pH 7. Figure 8 reports a comparison of 1D ^1^H and a selection of ^1^H-^1^H TOCSY spectra at pH 7 between the free peptide P23-42 and the Zn-bound system. 

The main perturbations were detected for Glu37 and Asp30, followed by Asp38. Moreover, a few non-coordinating neighboring residues underwent changes in the values of chemical shifts, i.e., Phe28, Lys31, Leu39, and Phe40. Small variations were also detected for the nuclei of His34. In particular, with the increase in the pH, it becomes clear how the imidazole protons of histidine are increasingly shifted, in accordance with the involvement of the imidazole ring in the binding with the metal ion at more basic conditions (Appendix A). From an NMR analysis, the behavior of the P23-42 peptide towards Zn^2+^ is similar to that observed for the shorter sequence P29-38. The anchoring site is centered on Glu-37 and Asp-38 and when the pH increases, the involvement of His34 occurs. Moreover, together with Asp30, Glu23 could also play a secondary role in the metal interaction since its nuclei experienced selective perturbations. Peptide P19-42 was revealed to be even less soluble than P23-42; however, we were able to extract some information from the NMR spectra. The longer peptide shows, broadly speaking, a coordination mode with Zn^2+^ very similar to that of the P23-42 peptide (Appendix A). The possibility of obtaining additional information at basic pH was precluded by the precipitation of the Zn system for both P23-42 and P19-42 peptides.

### 2.3. Copper(II) Complexes

The complex formation constants for the Cu(II)-P29-38 system are reported in Table 3, and the corresponding distribution diagrams are shown in Figure 9. UV–Vis and CD spectra at various pH values are given in Figure 10.

The formation of Cu(II) complexes with P29-38 begins at a pH of about 3.5 with the [CuH_2_L] species. Most likely, the metal is bound by the carboxylic moieties of the two aspartate and two glutamate residues. This species has a maximum concentration at pH 4.5, but does not reach the 30% of metal ions in solution. Consequently, due to its low abundance, the spectroscopic characteristics of this species are not accessible. Starting from pH 3.95, another proton is released, leading to the formation of the [CuHL]^−^ complex, which persists up to pH 7.5, with a maximum at pH 5.5. This proton derives from the deprotonation of the His34 imidazole that is able to coordinate the copper ion at this pH. In the UV−Vis spectrum recorded at pH 5.19, the presence of a band with a maximum absorption at 715 nm suggests a single nitrogen atom in the metal ion coordination sphere. The UV–Vis spectrum of [CuHL]^−^ calculated by SPECFIT/32 software [28] based on pH-dependent spectra (Figure 10 and Appendix A) shows a band characterized by λ_max_ = 720 nm and ε = 50 M^−1^·cm^−1^ (Table 4), consistent with the experimental spectrum and the literature values for 1N (N_Im_) Cu(II) d–d transition bands [29,30].

From CD spectroscopy it is possible to detect ellipticity starting from pH 5.19, with a positive absorption band appearing in the range of 500–600 nm, which increases up to pH 7.40, attributable to the interaction between His34 (ND2) and Cu^2+^, as well as the presence of characteristic CD bands at 235 and 340 nm (N_Im_ → Cu(II) MLCT).

Due to the paramagnetic character of copper, NMR experiments for the Cu(II)-peptide system were performed by progressive addition of substoichiometric amounts of metal ions to the peptide solutions to avoid severe broadening of the signals. By following the selective relaxation effect experienced by nuclei closest to the paramagnetic Cu(II), it is possible to localize in the amino acid sequence of the metal binding donors in the amino acid sequence and the changes undergone by the peptide upon interaction with the metal [26]. The NMR titration of P29-38 at pH 5.5 with increasing addition of sub stoichiometric amounts of Cu^2+^ is shown in Figure 11. In agreement with the proposed coordination mode {Asp, Asp, (Glu), His}, the analysis of the NMR spectra confirms the involvement of the imidazole nitrogen of His34 and the carboxylate of Asp30, Asp38, and Glu37, since their signals were selectively and progressively reduced during Cu(II) titration. The comparison of ^1^H-^13^C HSQC for the free P29-38 and for Cu(II)-P29-38 system in the molar ratio of 0.01:1 and 0.1:1 showed, moreover, all the residues that were excluded from the relaxation effect of Cu(II), namely Leu29, Lys31 and Ala36, giving an indirect indication of their major distance from the paramagnetic ion (Appendix A).

The next species [CuLH_-1_]^3−^ appears starting from pH 5 and derives from two consecutives deprotonations that can be assigned to two backbone amide nitrogens. The average corresponding pK_step_ value of 6.33 is compatible with this hypothesis. In fact, once anchored to the peptide through a His residue, Cu^2+^ is able to displace the amide protons through a cooperative effect, which facilitates the coordination of other amide nitrogens as the pH increases. Starting from pH 5.93, the UV–Vis absorption band begins shifting towards a shorter wavelength, thus reflecting the formation of this new complex form, [CuLH_-1_]^3−^, derived from two consecutives backbone amide nitrogen deprotonations. In this form, the coordination of the copper ion is most probably of the 3N {N_Im_, 2N^−^} type, λ_max_ = 622 nm and ε = 52 M^−1^·cm^−1^ (Table 4). The presence of [CuLH_-1_]^3−^ is also reflected in the CD spectra by the formation of the bands at 540 nm (d–d transition) and 310 nm (N^−^ → Cu(II) charge transfer transition), indicating the involvement of amides in metal coordination [31].

The NMR titration of P29-38 with increasing addition of substoichiometric amounts of Cu^2+^ was also performed at pH 7.4. Figure 12 and Appendix A report the results of this NMR study.

The NMR study carried out at pH 7.4 still clearly evidenced the involvement of His34 in the coordination sphere of Cu(II). The imidazole protons HD2 and HE1 experienced a severe line broadening, together with HB2-HB3 in its sidechain. Other residues influenced were the adjacent Asn33 and Phe32 (in particular their alpha and beta nuclei), which experienced a signal disappearance as an effect of their proximity to the paramagnetic metal ion. It is evident from the spectra that the participation of acidic residues in complex formation diminishes in accordance with the coordination mode change involving nitrogen amides. As the pH increases, another backbone amide deprotonates, with the p*K* value of 7.52. It is the [CuLH_-2_]^4−^ form, characterized by a 4N {N_Im_, 3N^−^} coordination. Due to the equilibrium in solution of the 3N and 4N coordination forms, it is difficult to clearly discern between their spectroscopic bands. In general, the 3N Cu(II) coordination in [CuLH_-1_]^3−^ form can be clearly seen at pH = 6.25, with λ_max_ = 622 nm. At a higher pH, up to 7.70, the band moves to a shorter wavelength of about 608 nm and its intensity rises as the third amide enters the coordination sphere in [CuLH_-2_]^4−^. The bands calculated by SPECFIT/32 are characterized by λ_max_ = 622 nm (ε = 52 M^−1^·cm^−1^) and λ_max_ = 608 nm (ε = 94 M^−1^·cm^−1^) for the 3N and 4N forms, respectively (Table 4 and Appendix A). A similar behavior is also reflected in the CD spectra, with the increasing intensity of the band at about 540 nm, rising up to pH = 7.60.

The next complex form, [CuLH_-3_]^5−^, is present in the solution already from a pH of about 7, with a maximum concentration at a pH of about 10. This form derives from the deprotonation of the fourth backbone amide nitrogen, with the p*K* value of 8.92. The imidazole nitrogen, bound to the Cu(II) ion already from a pH of about 4, is most probably displaced by the amide nitrogen, resulting in a 4N {4N^−^} coordination mode. This can be clearly seen in both the UV–Vis and CD spectra. In the UV–Vis spectrum recorded at pH = 8.07, another component of the band starts rising at about 520 nm, reflecting the equilibrium in solution between the [CuLH_-2_]^4−^, 4N {N_Im_, 3N^−^}, and [CuLH_-3_]^5−^, 4N {4N^−^}, forms. The UV–Vis spectrum calculated for [CuLH_-3_]^5−^ with λ_max_ = 520 nm (ε = 120 M^−1^·cm^−1^) (Table 4 and Appendix A) reflects the presence of a 4N^−^ form, consistent with the literature values [29,32]. In the CD spectrum recorded at the same pH, a drastic change compared to the previous spectra can be seen, with the appearance of new bands at 500 and 650, indicating the altered coordination of the metal ion. Indeed, these d–d transition bands are characteristic of the square planar, amide-involving complexes [33]. With the increasing pH up to about 11, the intensity of these bands increases as well, reflecting the rising concentration of the [CuLH_-3_]^5−^ form, shifting the equilibrium towards the 4N^−^ coordination mode. The last complex form, [CuLH_-4_]^6−^, is a result of the deprotonation of the non-binding ε-amino sidechain group of the lysine residue, with a p*K* value of 11.20.

Since at high pH values in the Cu(II) systems we detected a severe line broadening in the NMR spectra, which prevented a detailed characterization of the species in the [N_Im_, 3N^−^_amide_] coordination mode, we decided to probe Cu^2+^ binding to the P29-38 peptide using diamagnetic Ni^2+^. In terminally blocked peptides containing a histidine residue from the third position onwards, Ni^2+^ forms diamagnetic, low-spin, square planar complexes at appropriately high pHs. The imidazole nitrogen of histidine serves as an anchoring site and once bound to it, Ni^2+^ is able (like Cu^2+^) to deprotonate and bind three additional amide nitrogens from the backbone, thus completing the coordination sphere of [N_Im_, 3N^−^_amide_], with simultaneous formation of three fused chelated rings and the saturation of the coordination plane [34,35]. Therefore, Ni^2+^ could be used as the Cu^2+^ probe since it is able to replace it in the equivalent square planar complexes, but forming low-spin diamagnetic species which are more easily investigated by NMR [36,37]. Figure 13 reports the metal titration at pH 10.6, by addition of increasing amounts of Ni(II) ions, up to a 0.8 : 1, Ni(II):P29-38 molar ratio. Figure 14 shows the histograms related to the ^1^H and ^13^C chemical shift variations, the structural scheme of the peptide P29-38 with the most perturbed H and C nuclei, and the structural model proposed for the corresponding Ni^2+^ species.

Several changes were detected in the NMR signals of the peptide upon nickel addition. Diamagnetic complex formation is confirmed by the shift in numerous signals related to the specific residues His34, Asn33, Phe32, and Lys31. The involvement of His34 is immediately indicated by the gradual disappearing of the relative resonances of the imidazole HD2 and HE1 and aliphatic HA and HB signals in the free peptide and the simultaneous appearance of a new set of peaks related to the metal bound form in a slow exchange on the chemical shift timescale. In particular, the chemical shift differences (Δδ) between the imidazole aromatic proton from the bound and free state, greater for HE1 with respect to HD2 (Δδ HE1 = 0.783 ppm > Δδ HD2 = 0.121 ppm), give a clear indication of the metal bound to the adjacent imidazole nitrogen ND1 (Figure 13a and Figure 14a). All the His34 protons are shielded according to the increased electron density. The chemical shift changes are more pronounced for the residues taking part in metal coordination: His34, Asn33, Phe32, and Lys31. The major variations, according to the formation of a low-spin (square-planar or five-coordinate square-pyramidal) diamagnetic complex, are related to the HA protons, which are greatly shielded due to the influence of an increased electron density upon amide deprotonation (Figure 13b and Figure 14). The trend in changes in Δδ HA > HB and Δδ CA > CB is coherent with a coordination mode involving the backbone amide nitrogens of His34, Asn33, and Phe32. The HB2 and HB3 protons of Phe32 and Asn33 exhibit different electronic environments, as one is up-field and the other is down-field shifted. This behavior is most likely due to a blocked position of the Phe32 and Asn33 sidechains above the complex, which causes only part of the sidechain to be shielded (Figure 13c and Figure 14). ROE cross-correlations have in fact been identified between the phenolic HD proton of Phe32 and the imidazole HE1 proton of His34. Additional ROEs have been identified between neighboring nuclei and in particular between the sidechain protons of Phe32 and Asn33, which confirms their relative position on the same side of the coordination plane. Similarly, the sidechain protons of Lys31 are strongly deshielded due to a blocked conformation on the opposite side with respect to Asn33 and Phe32 sidechains. A more rigid spatial conformation of Lys31, under the plane of the complex, is confirmed by the fact that during Ni^2+^ titration the amide proton HN of Lys31 showed ROE correlations with the HA proton of Asp30 and the HA proton showed ROE correlations with the HE1 and HD2 protons of His34. Interestingly, this labile proton HN reappears after the addition of Ni^2+^, and it is probably blocked, as if it were involved in a hydrogen bond. The blocked conformation of the backbone relate to the position of Asp30 could explain, moreover, the possibility of a five-coordinate square-pyramidal coordination through the carbonyl of Asp30 interacting with the metal ion in an axial position under the plane of the complex, as suggested by the model depicted in Figure 14c. To verify whether Ni^2+^ was a good probe of Cu^2+^, we performed NMR competition studies by titrating the diamagnetic Ni(II):P29-38 system (0.8:1 molar ratio) with increasing substoichiometric amounts of paramagnetic Cu^2+^. The already shifted signals due to Ni(II) binding, related to the residues taking part in metal complexation, selectively disappeared during the titration, an indication that Cu^2+^ ions were able to displace and substitute Ni^2+^ ions in the same coordination site (Appendix A). Moreover, the addition of Cu^2+^ to the Ni(II):P29-38 system showed further clues about the involvement of the fourth amide N^−^ in the complexation, since the already cited amide HN signal of Lys31 disappears due to its deprotonation and subsequent coordination with the metal ion. The involvement of Lys31 is furthermore evidenced by the disappearing or by the severe lowering of its spin system signals, as an indication of its new location in the proximity of the paramagnetic ion. A proposed structural model for Cu(II):P29-38 in a 4N^−^_amide_ coordination sphere is depicted in Figure 15.

The NMR study of the longer peptides P23-42 and P19-42 provided some evidence on the coordination preferences of Cu^2+^. At pH 7, both Cu(II)-P23-42 and Cu(II)-P19-42 systems showed a selective disappearance of the signals related to His-34, Asp-30, and Asp-38 (Appendix A) and when the pH was set to 8, the involvement of Glu-35 and Glu-37 appeared more clearly. At higher pHs, the concomitant effect of excessive line-broadening and poor solubility prevented a better characterization of Cu(II) systems for both the P23-42 and P19-42 peptides. However, to obtain further information on the species forming at high pH values, we used Ni^2+^ as a Cu^2+^ probe also for the longer peptide P19-42. The results are strikingly similar to those obtained for the shorter peptide P29-38. In fact, the titration at pH 10.6 led to a system in which Ni forms a diamagnetic planar complex involving, similarly to peptide P29-38, not only the same coordinating residues of His34 (N_im_, N^−^), Asn33 (N^−^), and Phe32 (N^−^) (Figure 16 and Appendix A), but also showing almost identical chemical shift differences (Δδ ppm) between the two systems (Figure 17). As in the case of minimal model P29-38, the labile HN proton of Lys31 reappears after addition of Ni^2+^ and also several ROEs that were identified for the Ni^2+^-P29-38 system were confirmed for Ni^2+^-P19-42 system, in particular Asp30HA-Lys31HN, Lys31HA-His34HD2, Lys31HA-His34HE1, Phe32HB2-His34HD2, Asn33HB2-Phe32HD, Asn33HB3-Phe32HD, and His34HE1-Phe32HD, which were related to the blocked conformation of sidechain Phe32 and Asn33 above and Lys31 under the plane of the complex.

A competition study was also conducted for the longest peptide with increasing amounts of Cu^2+^ added to the Ni(II)-P19-42 system at pH 10.6. Exactly, as in the case of the shorter peptide P29-38, the signals related to Ni^2+^ binding and involving His34, Asn33, and Phe32, which play a main role in metal complexation, selectively disappeared following the substitution of the paramagnetic Cu^2+^ ion with the diamagnetic Ni^2+^ in an identical coordination sphere around the metal ion (Appendix A). The involvement of the amide nitrogen of Lys31 in the 4N^−^ coordination mode is evidenced, as it was before, by the disappearance of its HN spin system.

### 2.4. Competition Diagrams

The competition plot between P29-38 and Zn(II) and Cu(II) ions illustrates (Figure 18) the complex formation in a hypothetical situation, when equimolar amounts of all reagents are mixed in the solution. Up to a pH of about 2.8, almost all of the ligand exists in an uncomplexed form. From pH = 3, the beginning of complexation can be clearly seen, with copper complexes dominating over zinc complexes already in these acidic conditions, a behavior which persists all the way up to pH = 11. Cu(II) complexes quickly reach over 60% of the ligand molecules bound at a pH of about 4. Around this pH, the maximum concentration of Zn(II) complexes is also achieved, reflecting the start of deprotonation and binding of histidine H34 by Zn^2+^. As the pH increases, the Zn(II) complex concentration starts to drop, with less than 5% percent of the ligand bound at pH = 7. On the contrary, a rise in the Cu(II) complex concentration can be seen, reflecting the increasing involvement of nitrogen ligands in copper binding, such as the histidine residue and amide groups. Reaching 90% of the ligand bound already at pH = 6.50, Cu(II) complexes clearly dominate over those of Zn(II), whose coordination sphere remains unchanged in the basic pH range. The behavior observed in the competition plot of P29-38 with Zn(II) and Cu(II) is in agreement with the Irving–Williams series, in which the stability of divalent copper complexes is higher than those of divalent zinc complexes [38]. The results of this experiment remain valid for a hypothetical situation where equimolar amounts of all reagents are used. In human cells, the situation can be completely reversed, since the relative concentrations of free copper and zinc in plasma are not the same, being about 10^−18^–10^−13^ M for copper and around 10^−9^ for zinc [39,40].

To explore the stability of the P29-38 complexes in more detail, we calculated the dissociation constant K_d_, which refers to the concentration of the free metal ion (expressed in molarity) when half of the ligand exists in a complex form and the other half is not complexed [41]. K_d_ does not depend on the ligand concentration, although it depends on the pH. Since K_d_ refers to the general equilibrium ML = M+ L, the lower the value of the constant, the greater the stability of the complex [42,43]. The values of dissociation constants are widely used to compare the stability of the complexes of endogenous ligands. We compared the K_d_ values obtained at the physiological pH of 7.4 for our system with those of important endogenous ligands: human serum albumin (HSA) for Cu(II) and Zn(II) and reduced glutathione (GSH) for Zn(II). The results are reported in Table 5. 

The dissociation constant value for Zn(II) complex of P29-38 is higher than those of HSA and GSH, which means that they indeed generate more stable complexes with Zn(II) than P29-P38. However, with the K_d_ value for our system being only about 100 times higher than those of very efficient endogenous zinc ligands, the P29-P38 ligand most likely could effectively bind zinc also in vivo. The K_d_ value for the P29-38 Cu(II) complex is about 10,000 times higher than that for HSA, marking a greater difference than that observed for Zn(II); however, it is known that the ATCUN motif present in the N site-terminus of albumin is a very efficient binding site for Cu(II).

## 3. Materials and Methods 

### 3.1. Peptide Synthesis

P29-38 (Ac-LDKFNHEAED-NH_2_), P23-42 (Ac-EQAKTFL_29_DKFNHEAEDLFYQ-NH_2_), and P19-42 (Ac-STIEEQAKTFLDKFNHEAEDLFYQ-NH_2_) peptides were purchased from Biomatik Corporation, Kitchener, Ontario, Canada. All peptides have a purity > 97%. All of the chemicals used in this work were purchased from Sigma-Aldrich (St. Louis, MO, USA) and used without any further purification.

### 3.2. Potentiometric Measurements

All the potentiometric data were calculated from two titration experiments carried out over the pH range 2.0–11.0 at 298 K in 0.1 M NaClO_4_ using a total volume of 2 mL. A Metrohm Titrando 905 titrator connected to a Dosino 800 dosing system and a pH electrode (InLab Semi-Micro (Mettler-Toledo)) were used to carry out the experiments. The electrode was calibrated every day for hydrogen ion concentration by titrating 2 mL of 4 mM perchloric acid with sodium hydroxide. All potentiometric measurements were performed under an argon atmosphere. Purities and exact concentrations of ligand solutions were determined by the Gran method [46]. The ligand concentration was 0.5 mM and the metal to ligand molar ratio was 1:1.1. HYPERQUAD 2008 [47] and SUPERQUAD [48] programs were used to calculate the stability constants. Standard deviations were computed using HYPERQUAD 2008 and they refer to random errors only. Cu(II) and Zn(II) hydrolysis constants were taken into account for the calculations of stability constants of complexes. The hydrolysis constants for zero ionic strength were taken from the “*Hydrolysis of Metal Cations*” by Brown and Ekberg [49] and calculated to 0.1 M ionic strength with the formula proposed by Baes and Mesmer in “*The Hydrolysis of Cations*” [50]. The metal hydrolysis constants are collected in Appendix A. The competition and speciation diagrams were created using HYSS software [51].

### 3.3. UV-Vis and CD Measurements

Absorption spectra were recorded in the 240–800 nm range on a Jasco J-1500 spectropolarimeter (CD) and on a Jasco V-750 spectrophotometer (UV–Vis); solutions were of similar concentrations to those used in the potentiometric studies. The ligand concentration was 0.5 × 10^−3^ mol dm^−3^ and the tested Cu(II)/ligand molar ratio was 1:1. To calculate the absorptivities (ε, M^−1^·cm^−1^) of the various Cu(II) complex forms, as calculated by potentiometry and shown in distribution diagrams, UV−Vis data were refined using SPECFIT/32 software that adjusts the absorptivity and the stability constants of the species formed at equilibrium. As the d–d bands of Cu(II) complexes are rather broad and weak and the complexes are not very well separated, we fixed the stability constants calculated from potentiometric data and calculated only the spectra. SPECFIT uses a factor analysis to reduce the absorbance matrix and to extract the eigenvalues prior to the multiwavelength fit of the reduced dataset according to the Marquardt algorithm [28,52].

### 3.4. NMR Measurements

NMR experiments were performed using a Bruker Ascend™ 400 MHz spectrometer equipped with 5 mm automated tuning and a matching broad band probe (BBFO) with z-gradients. Samples used for NMR experiments were in the range 0.4–2.0 mM and dissolved in 90/10 (*v*/*v*) H_2_O–D_2_O. All NMR experiments were performed at 298 K in 5 mm NMR tubes. The 2D ^1^H–^13^C heteronuclear correlation spectra (HSQC) were acquired using a phase-sensitive sequence employing Echo-Antiecho-TPPI gradient selection with a heteronuclear coupling constant J_XH_ = 145 Hz and shaped pulses for all 180° pulses on the f2 channel with decoupling during acquisition. Sensitivity improvement and gradients in back-inept were also used. Relaxation delays of 2 s and 90° pulses of about 10 μs were applied for all experiments. Solvent suppression for ^1^H and ^1^H–^1^H TOCSY experiments was achieved using excitation sculpting with gradients. The spin-lock mixing time of the TOCSY experiment was obtained with MLEV17. ^1^H–^1^H TOCSY experiments were performed using mixing times of 60 ms. ^1^H–^1^H ROESY spectra were acquired with spin-lock pulses duration in the range 200–250 ms. The assignments of ^1^H and ^13^C were made by a combination of mono- and bi-dimensional and multinuclear NMR techniques, i.e., ^1^H–^1^H TOCSY, ^1^H–^13^C HSQC, and ^1^H–^1^H ROESY, at different pH values. All NMR data were processed using TopSpin (Bruker Instruments) software and analyzed using Sparky 3.11 and MestReNova 6.0.2 (Mestrelab Research S.L.) programs.

### 3.5. Molecular Dynamics Measurements

Extended peptide conformers were generated in Avogadro. Energy minimization was achieved using a restricted Hartree–Fock SCF, calculation was performed using Pulay DIIS + Geometric Direct Minimization and the 3-21G(*) basis set using Gaussian 16.

Subsequent calculations were carried out using available NAMD 2.14 and VMD 1.9.3 software. Structures were parameterized in CHARMM-GUI using the CHARMM36m force field. A rectangular waterbox was used to simulate the solvent behavior, sized as the protein size, and completed with 0.05 M KCl ions to balance the charge of deprotonated amino acids. Ions were placed using the Monte-Carlo method. The simulations were carried out at a temperature of 298 K. Each simulation was executed for 1 μs (following an initial minimization) with structures calculated every 10 ps and written to a trajectory file. For the 1 μs simulation, 500,000 time steps were recorded (2 fs = step size). Following the calculations, RMSD data plots relative to the extended, minimized initial structure were generated from the trajectory.

Model calculations for the Ni(II)-P29-38 complex were performed on the basis of the experimental evidence and the identification of specific ROE cross-correlations extracted from 2D ^1^H-^1^H ROESY spectra for the metal-bound system. At the pH value investigated, no signals were detected in the aromatic region for the labile amide protons H_N_, except for the Phe32 and His34 ring protons. ROE cross-peaks for the Ni(II)-P29-38 system at a 0.8:1 molar ratio were assigned, and the intensities were converted into the maximal distances. Upper bounds *u* of the distance between two correlated hydrogen atoms were derived from the corresponding ROESY cross-peak volumes *V* according to calibration curves *V = k/u^6^*, with the constant *k* determined by using the cross-peak intensity of a selected H–H cross-peak between nuclei with a known distance [37]. The structures shown in the RMSD trajectories were visualized and extracted using VMD and Chimera [53].

## 4. Conclusions

Zinc is an important cofactor for stabilizing protein structures and altering the substrate affinity of various metalloproteins. Zn^2+^ homeostasis could affect ACE2 expression, and binding to its active site is essential for its enzymatic activity [17]. It is very probable that Zn^2+^ binding to ACE2 could influence the molecular structure of the receptor and therefore its binding affinity with SARS-CoV-2. At the same time, a large number of zinc-deficient patients are more prone to developing severe COVID-19 [54,55]. The work presented here provides valuable insights into the complex coordination chemistry of Zn(II) and Cu(II) with three peptide fragments from the C-terminus of the ACE-2 receptor, a region that plays a crucial role in the binding with the S protein of SARS-CoV-2. This study utilized a combination of potentiometry, UV–Vis and CD spectroscopy, and NMR techniques to investigate the formation and coordination modes of Zn(II) and Cu(II) ions with selected peptide fragments that mimic the interface recognition region of ACE2 for S. A Ni(II) diamagnetic analog was also studied as a Cu(II) probe, since at high pH values the Cu(II) systems experienced a severe line broadening in the NMR spectra, which prevented detailed characterization of the species in the [N_Im_, 3N^−^_amide_] coordination mode. The results confirmed the bioinformatic previsions obtained by MIB2, which evidenced a highly specific region in ACE2 for zinc binding in the D30-Glu37 region. All experimental data indicate that this sequence is able to bind both metal ions in a very selective way. Moreover, the two long peptides mimicking the whole domain show the same coordination patterns as the shorter one, indicating the latter is an excellent model to study the coordinating abilities at the ACE2/S interface. Both metal ions can change the peptide conformation, which rearranges itself upon coordination with zinc and copper. Such structural modifications might interfere with the recognition mechanism of ACE2 and the S protein, also considering that some of the residues taking part in or being perturbed by metal coordination play a crucial role in hydrogen bond formation between the two proteins. These structural changes may decrease the affinity between ACE2 and the S protein and positively interfere in the mechanism through which the virus enters the cells. Therapies targeting ACE2 provide a general strategy to prevent and treat infections by SARS-CoV-2 and its variants, as well as other potential coronaviruses that use the ACE2 receptor as an entry route for viral invasion [56]. Whether it is possible to exploit this information in order to arrange a therapeutic strategy could be the topic of new research and opens new perspectives in the treatment of COVID-19 in those patients who still experience severe symptoms after infection, or in case new aggressive variants emerge. 

## Figures and Tables

**Figure 1 ijms-24-09202-f001:**
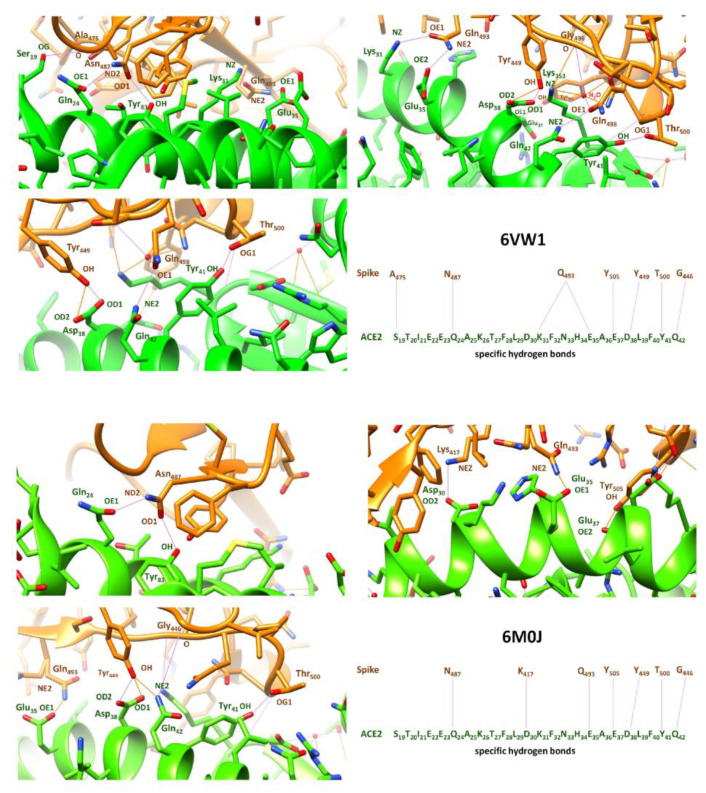
Specific hydrogen bonds between the RBD of SARS-CoV-2 and the binding interface of ACE2, extracted from the X-ray structures with PDB code 6VW1 [13] and 6M0J [12].

**Figure 2 ijms-24-09202-f002:**
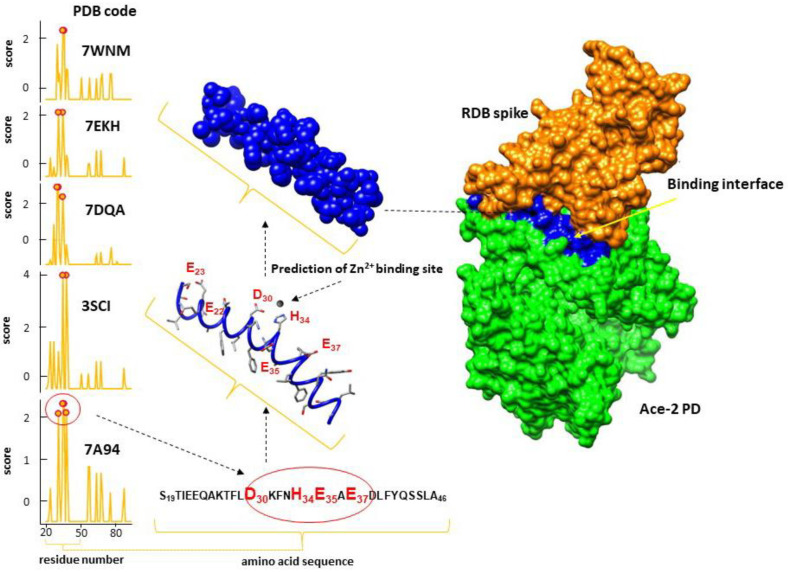
MIB2 prediction of Zn-ion-binding sites (in the range 19 to 100) from a selection of ACE2 X-ray structures (PDB 7A94, 3SCI, 7DQA, 7EKH, and 7WNM) [21,22,23,24,25]. MIB2 combines both structural and sequence information to identify the local structure of protein–metal interaction sites. The plot on the left shows the residues scoring above the threshold that are predicted to be metal-binding residues. The residues recurrent in all predictions are located in the binding interface between ACE2 and the S protein, localized in the fragment 30–37 aa.

**Figure 3 ijms-24-09202-f003:**
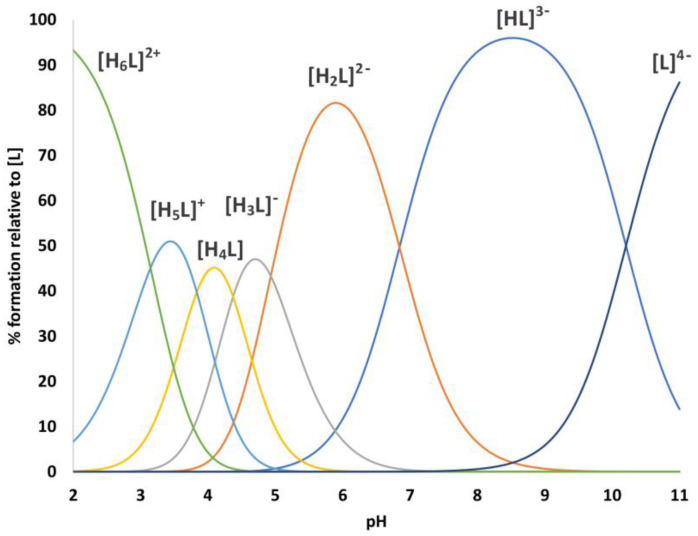
Representative distribution diagram for P29-38 peptide.

**Figure 4 ijms-24-09202-f004:**
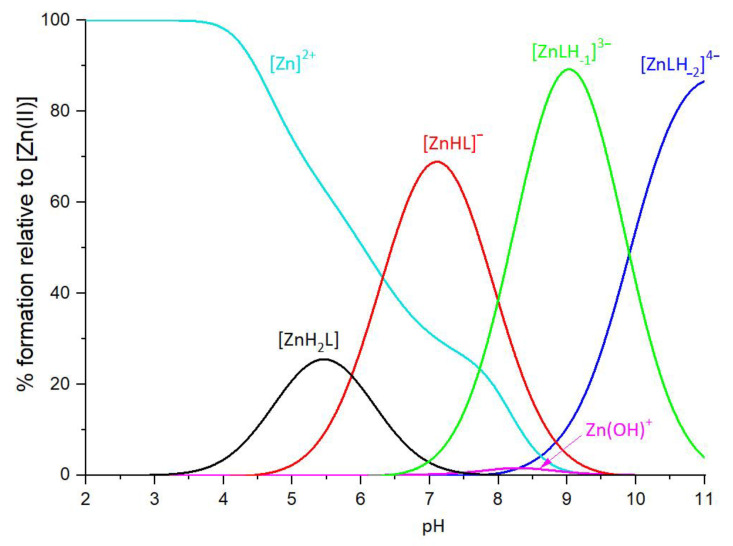
Representative distribution diagrams. [Zn(II)]_tot_ = 0.4 mM; Zn(II)/L ratio =1:1.1.

**Figure 5 ijms-24-09202-f005:**
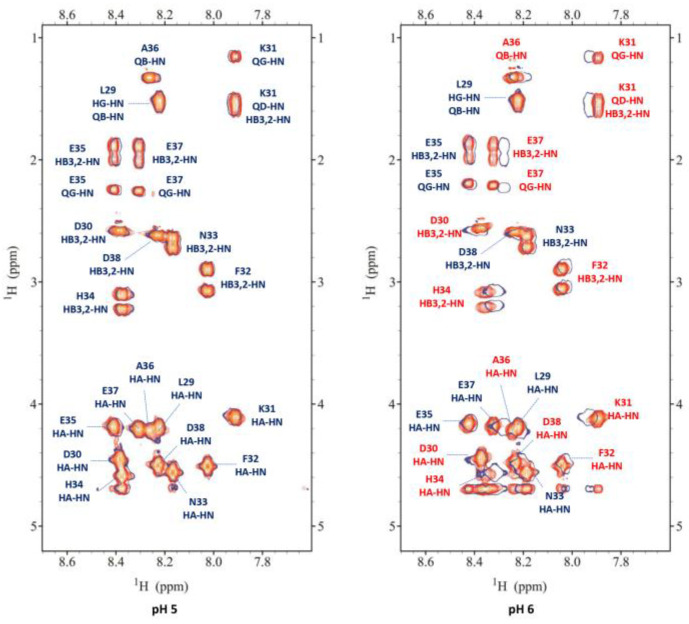
Superimposition of selected regions of the ^1^H-^1^H TOCSY spectra for the free peptide P29-38 (orange) and Zn(II)-P29-38 system (blue contours) at pH 5 and 6.

**Figure 6 ijms-24-09202-f006:**
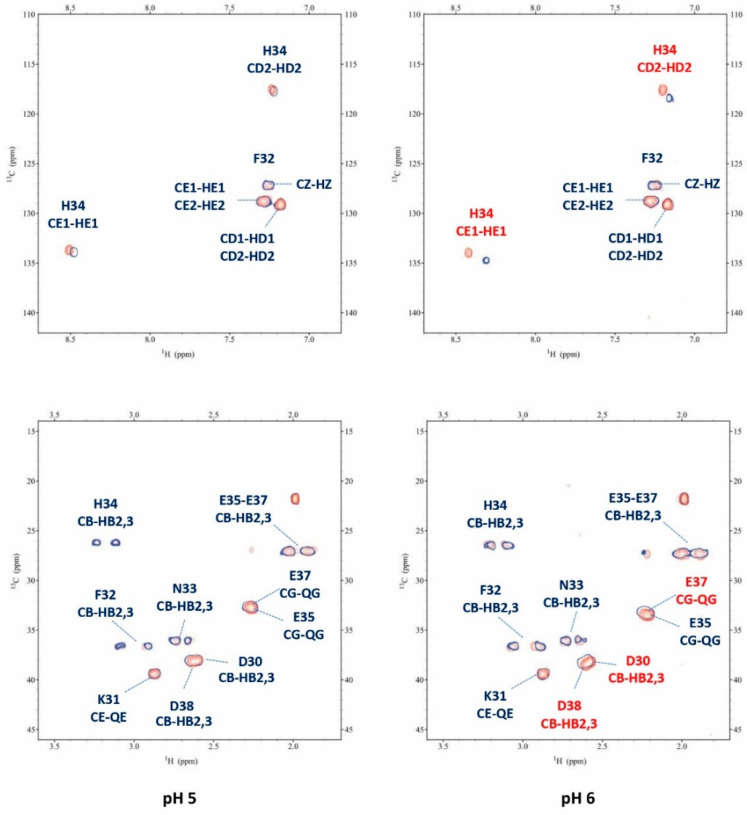
Superimposition of selected regions of the ^1^H-^13^C HSQC spectra for the free peptide P29-38 (orange) and Zn(II)-P29-38 system (blue contours) at pH 5 and 6.

**Figure 7 ijms-24-09202-f007:**
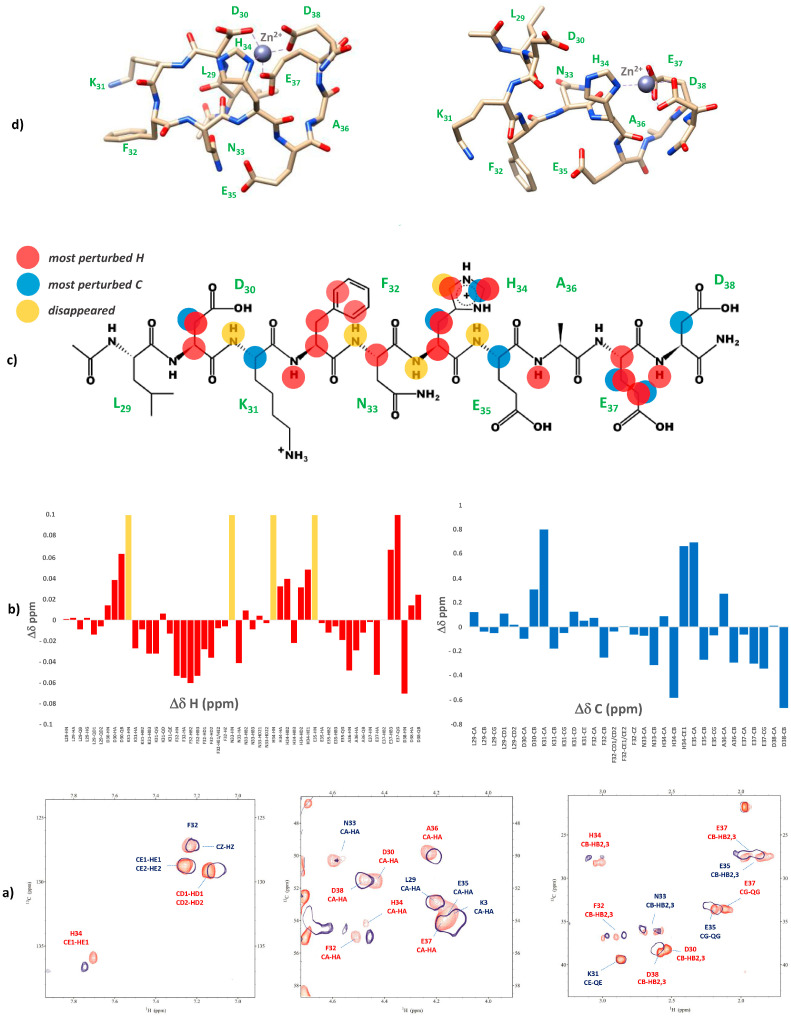
(**a**) Superimposition of selected regions of the ^1^H-^13^C HSQC spectra for the free peptide P29-38 (orange) and Zn(II)-P29-38 system (blue contours) at pH 7.4; (**b**) ^1^H and ^13^C chemical shift variations (Δδ = (Zn^2+^-P29-38 system)—(P29-38 free)); (**c**) structural scheme of the peptide P29-38 upon Zn^2+^ binding with the most perturbed H and C nuclei highlighted; (**d**) structural models of the [ZnHL]^−1^ species in a {(Asp), Asp, Glu, His} coordination mode.

**Figure 8 ijms-24-09202-f008:**
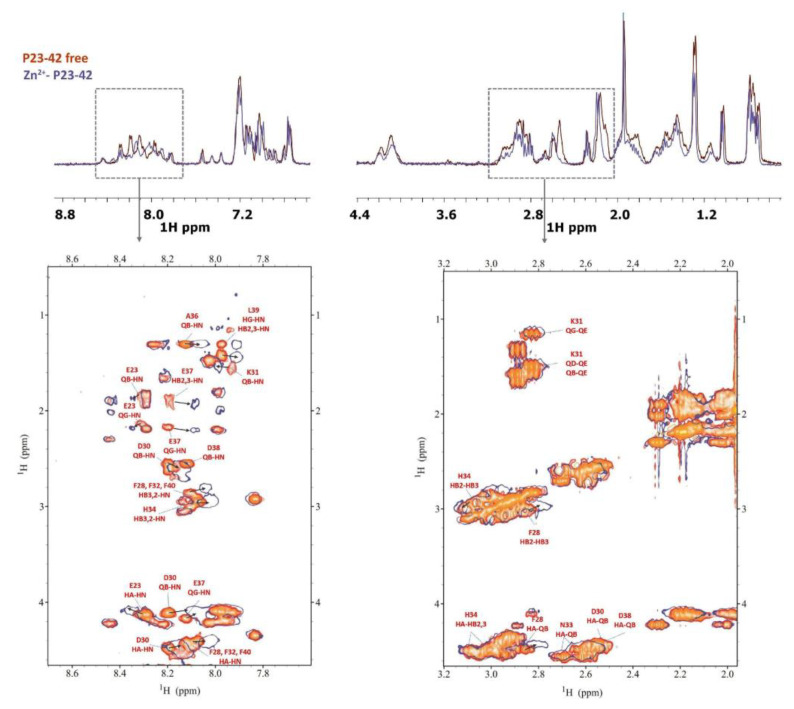
Comparison of ^1^H spectra (**up**) and selection of ^1^H-^1^H TOCSY spectra (**down**) for the free peptide P23-42 (orange) and Zn(II)-P23-42 system (blue) at pH 7.0.

**Figure 9 ijms-24-09202-f009:**
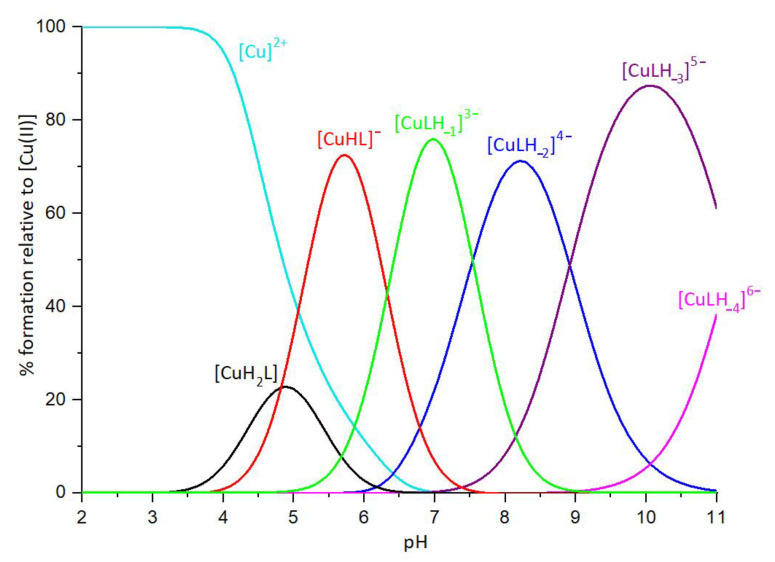
Representative distribution diagram. [Cu(II)]_tot_ = 0.4 mM; Cu(II)/L ratio = 1:1.1.

**Figure 10 ijms-24-09202-f010:**
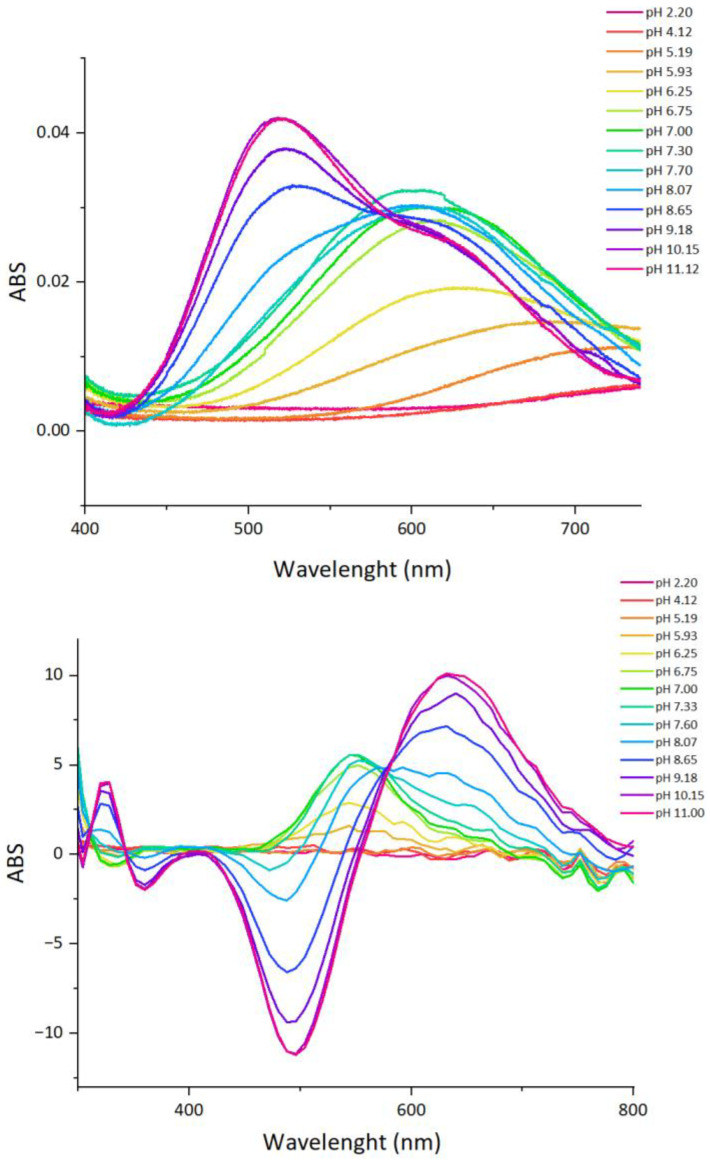
Cu(II)-P29-38 system. UV−vis spectra (**up**) and CD spectra (**down**) as a function of pH.

**Figure 11 ijms-24-09202-f011:**
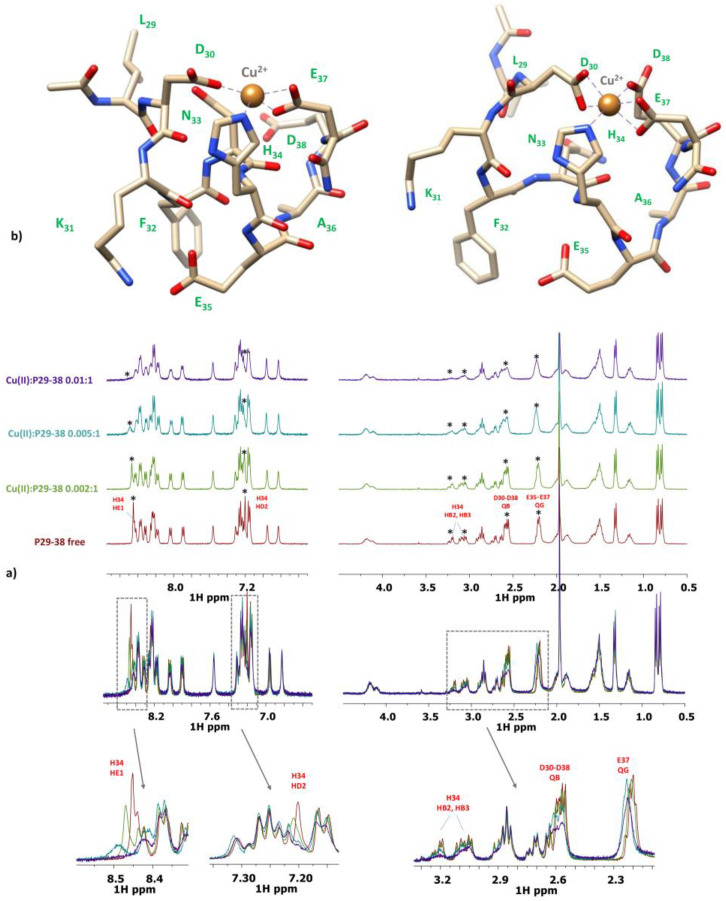
(**a**) Comparison of ^1^H spectra of P29-38 with increasing addition of Cu^2+^ at pH 5.5; * refers for the signals experiencing selective line-broadening; (**b**) Structural models of the [CuHL]^−1^ species in a {Asp, Asp, (Glu), His} coordination mode.

**Figure 12 ijms-24-09202-f012:**
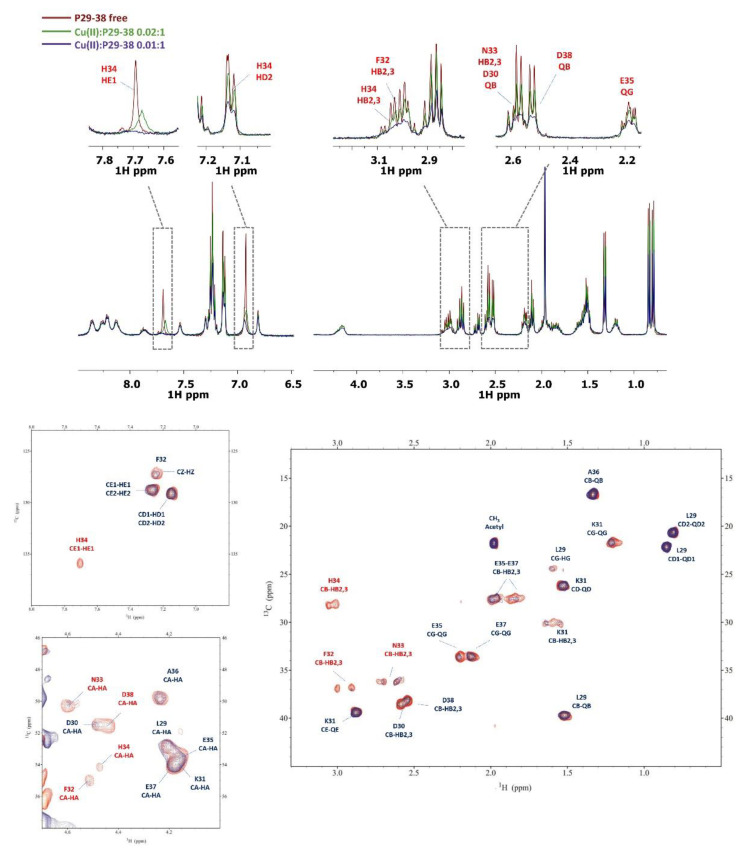
(**top**) Comparison of the ^1^H spectra of P29-38 with increasing addition of Cu^2+^ at pH 7.4; (**bottom**) comparison of the ^1^H-^13^C HSQC spectra of P29-38 free (orange) and Cu(II)-P29-38 at a 0.1:1 molar ratio (blue) at pH 7.4.

**Figure 13 ijms-24-09202-f013:**
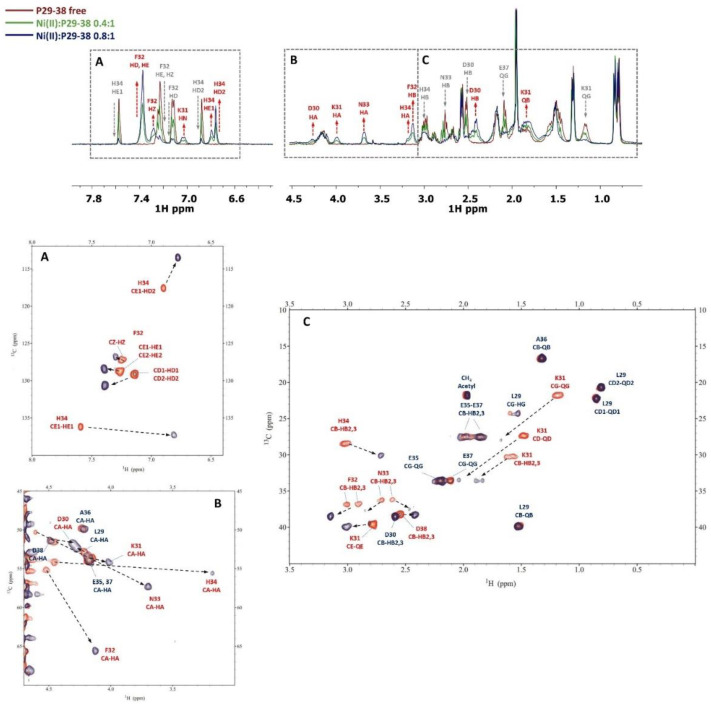
(**top**) Comparison of the ^1^H spectra of P29-38 with increasing addition of Ni^2+^ at pH 10.6. Gray arrows indicate signals that gradually decrease in intensity, while red arrows refer to new signals that gradually appear upon addition of Ni^2+^; (**bottom**) comparison of the ^1^H-^13^C HSQC spectra of P29-38 free (orange) and Ni(II)-P29-38 at a 0.8:1 molar ratio (blue) at pH 10.6; the black arrows indicate the new positions of the NMR signals following the addition of Ni^2+^.

**Figure 14 ijms-24-09202-f014:**
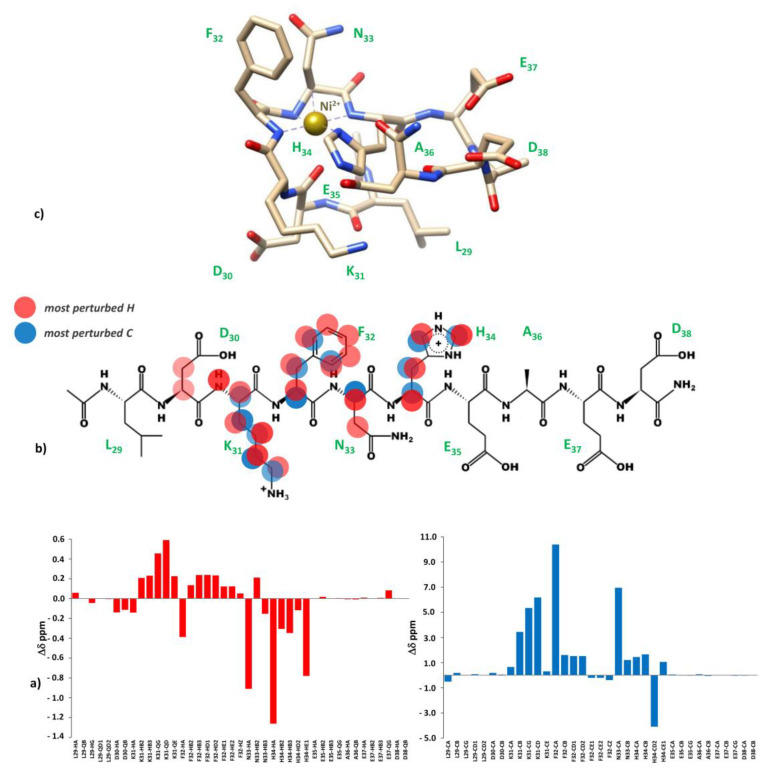
(**a**) ^1^H and ^13^C chemical shift variations [Δδ = (Ni^2+^-P29-38 system)—(P29-38 free)]; (**b**) structural scheme of the peptide P29-38 with the most perturbed H and C nuclei (highlighted in red and blue, respectively) upon Ni^2+^ binding; (**c**) Structural model of the Ni^2+^ species in a [N_Im_, 3N^−^_amide_] coordination mode.

**Figure 15 ijms-24-09202-f015:**
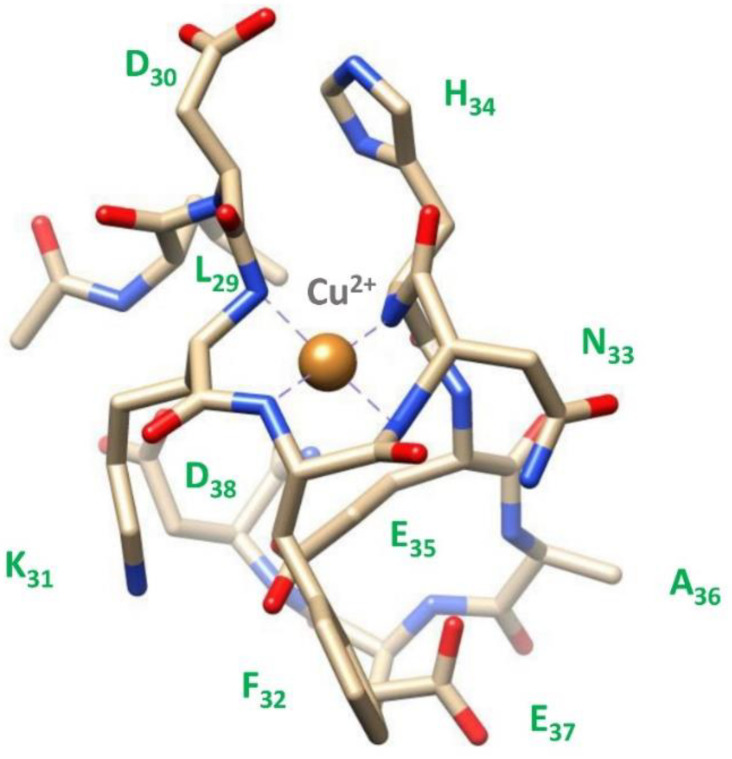
Structural model of the Cu^2+^ species in a [4N^−^_amide_] coordination mode.

**Figure 16 ijms-24-09202-f016:**
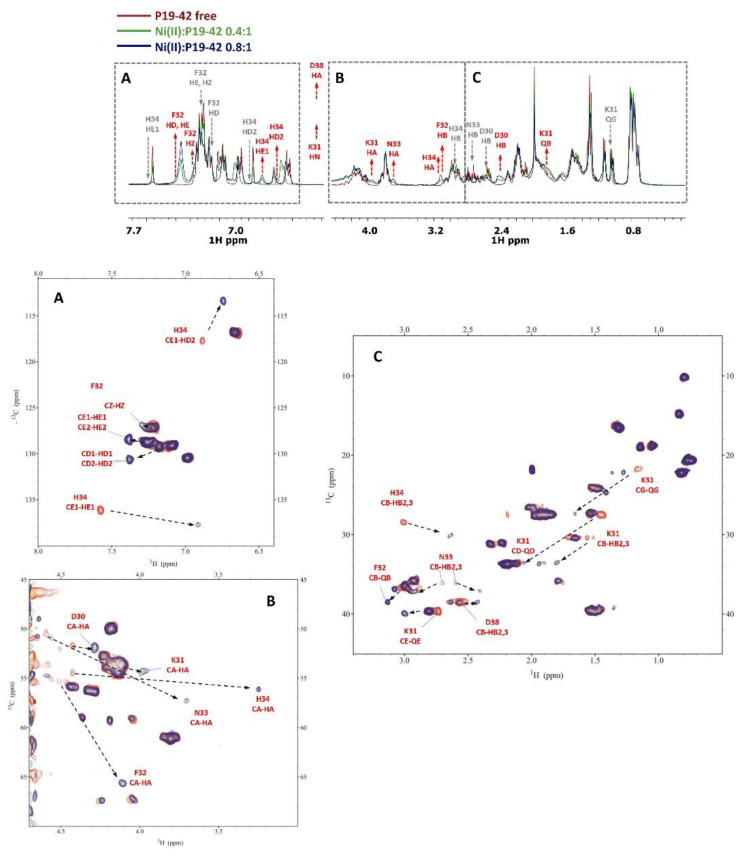
(**top**) Comparison of the ^1^H spectra of P19-42 with increasing amounts of Ni^2+^ at pH 10.6. The gray arrows indicate signals that gradually decrease in intensity, while the red arrows refer to new signals that gradually appear upon addition of Ni^2+^; (**bottom**) comparison of the ^1^H-^13^C HSQC spectra of P19-42 free (orange) and Ni(II)-P19-42 at a 0.8:1 molar ratio (blue) at pH 10.6; the black arrows indicate the new positions of the NMR signals following the addition of Ni^2+^.

**Figure 17 ijms-24-09202-f017:**
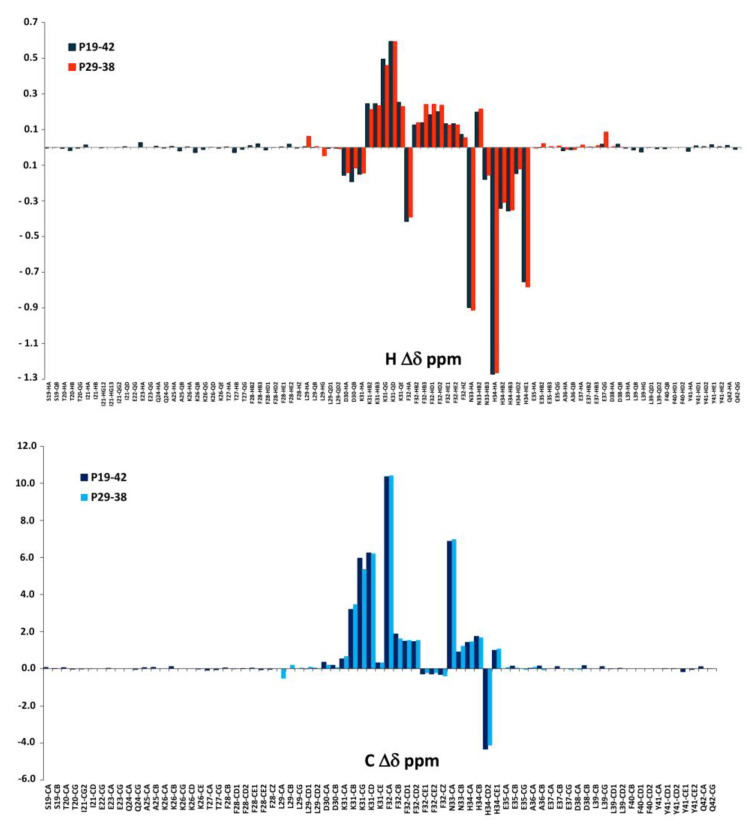
Comparison of ^1^H and ^13^C chemical shift variations for P19-42 and P29-38 peptides upon Ni^2+^ coordination.

**Figure 18 ijms-24-09202-f018:**
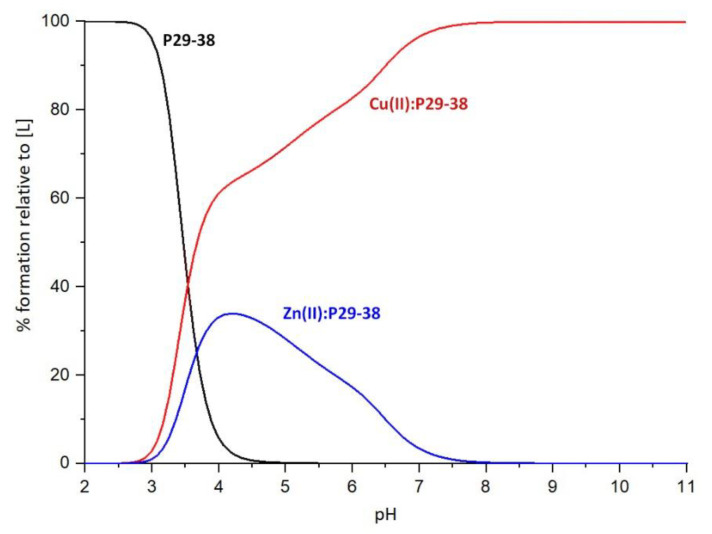
Competition plot between P29-38 and Zn(II) and Cu(II), depicting a hypothetical situation in which all reagents are mixed in equimolar concentrations.

**Table 1 ijms-24-09202-t001:** Protonation constants at 298 K and I = 0.10 mol dm^−3^, precision on the last significant figure in parentheses.

Ligand Species	Log*β*	log*K*	Residue
[HL]^3−^	10.21 (1)	10.21	Lys
[H_2_L]^2−^	17.05 (2)	6.84	His
[H_3_L]^−^	21.98 (2)	4.93	Glu
[H_4_L]	26.36 (2)	4.38	Glu
[H_5_L]^+^	30.19 (2)	3.83	Asp
[H_6_L]^2+^	33.34 (2)	3.15	Asp

**Table 2 ijms-24-09202-t002:** Complex formation constants for Zn(II) complexes at 298.2 K and I = 0.1 mol dm^−3^. The precision of the last significant figure is in parentheses.

Species	log*β*	log*K*	Deprotonation	Coordination
[ZnH_2_L]	20.34 (8)			Asp, Asp, Glu, (Glu)
[ZnHL]^−^	14.55 (6)	5.79	His	(Asp), Asp, Glu, His
[ZnLH_-1_]^3−^	−1.50 (5)	8.02 × 2	2H_2_O	(Asp), Asp, Glu, His
[ZnLH_-2_]^4−^	−11.42 (6)	9.92	Lys	(Asp), Asp, Glu, His

**Table 3 ijms-24-09202-t003:** Complex formation constants for Cu(II) complexes at 298.2 K and I = 0.1 mol dm^−3^. The precision of the last significant figure is shown in parentheses.

Species	log*β*	log*K*	Deprotonation	Coordination
[CuH_2_L]	20.78 (5)			Asp, Asp, Glu, Glu
[CuHL]^−^	16.01 (3)	4.77	His	Asp, Asp, (Glu), His
[CuLH_-1_]^3−^	3.35 (3)	2 × 6.33	amide, amide	His, 2N^−^_amide_, (Asp or Glu)
[CuLH_-2_]^4−^	−4.17 (4)	7.52	amide	His, 3N^−^_amide_
[CuLH_-3_]^5−^	−13.09 (5)	8.92	amide	4N^−^_amide_
[CuLH_-4_]^6−^	−24.29 (5)	11.20	Lys	4N^−^_amide_

**Table 4 ijms-24-09202-t004:** UV–Vis spectroscopic parameters of nitrogen-coordinated complex forms in the Cu(II)-P29-38 system calculated by SPECFIT/32 software.

Complex Form	Coordination	λ [nm]	ε [M^−1^·cm^−1^]
[CuHL]^−^	1N (N_Im_}	720	50
[CuLH_-1_]^3−^	3N {N_Im_, 2N^−^}	622	52
[CuLH_-2_]^4−^	4N {N_Im_, 3N^−^}	608	94
[CuLH_-3_]^5−^	4N {4N^−^}	520	120

**Table 5 ijms-24-09202-t005:** Dissociation constant (K_d_) values for the complexes of Cu(II) and Zn(II) with P29-38, human serum albumin and glutathione at pH = 7.4.

Ligand	K_d_ (Zn^2+^)	K_d_ (Cu^2+^)
P28-38	5.39 × 10^−5^	8.18 × 10^−9^
HSA ^a^	1 × 10^−7^	9.55 × 10^−14^
GSH ^b^	5.74 × 10^−7^	-

^a^ For Zn(II) the K_d_ value is for the complex with metal-binding site A of HSA, for Cu(II) the K_d_ value is for the complex with the N-terminal site [44]. ^b^ Protonation and stability constants used for K_d_ calculations taken from [45].

## Data Availability

All data generated or analyzed during this study are available under request.

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
