# Peer review of "Zn2+ and Cu2+ Interaction with the Recognition Interface of ACE2 for SARS-CoV-2 Spike Protein"

_ijms, 2023, doi:10.3390/ijms24119202_

Round 1

Reviewer 1 Report

This is an impressive paper.  The authors have made excellent use of potentiometry and spectroscopy to study metal complexation of the ACE2 protein, targeting the binding region of SARS-CoV-2.

The paper is well-written and easy to follow.  I particularly liked the use of Cu(II) aa a paramagnetic probe in the NMR experiments.  The findings are well supported by the experimental results.

A few minor points:

There are a few grammatical errors and typing mistakes eg 'trough' instead of 'though'.

In the text, the authors do not mention that DFT calculations were carried out - it is mentioned in the 'materials and methods' section.  Since DFT was done did the authors calculate the electronic transitions of the Cu complexes and if so how did they compare?

I am surprised no paramagnetic shifts were seen in the Ni spectra at pH values where the square planar, diamagnetic complexes are not formed.

At high pH we have often found that the Cu complex enters a region of slow exchange and so the NMR signals sharpen - ie the spectrum is of the free ligand.

The authors calculate the relative binding of Zn and Cu and find Cu is better.  This is not really relevant.  What is more important is how stable the Zn/peptide complexes are relative to other endogenous ligands.   

The measured equilibrium constants have quite low standard deviations, given the number of titrations performed.  Do we have any idea as to the 'goodness of fit' between the experimental and theoretical data.  

SPECFIT was used to calculate the electronic spectra, but the equilibrium constants were adjusted as well.  Why? 

Author Response

This is an impressive paper.  The authors have made excellent use of potentiometry and spectroscopy to study metal complexation of the ACE2 protein, targeting the binding region of SARS-CoV-2.

The paper is well-written and easy to follow.  I particularly liked the use of Cu(II) aa a paramagnetic probe in the NMR experiments.  The findings are well supported by the experimental results.

Authors reply:

We thank the reviewer for the positive comments and valuable suggestions. In the revised version, we made any effort to follow the suggestions in order to improve the quality of the MS.

A few minor points:

There are a few grammatical errors and typing mistakes eg 'trough' instead of 'though'.

Authors reply:

In the revised version of the MS the language has been carefully checked and grammatical errors and typos have been corrected

In the text, the authors do not mention that DFT calculations were carried out - it is mentioned in the 'materials and methods' section.  Since DFT was done did the authors calculate the electronic transitions of the Cu complexes and if so how did they compare?

Authors reply:

Thank you for your insightful comment. In our study, we elected to use the Hartree-Fock method with a 3-21G* basis set, along with molecular modelling calculations, to determine the structure of metal complexes. While we acknowledge that DFT calculations may provide different insights, particularly for electronic transitions, our choice of methodology was driven by certain considerations. These include computational efficiency, the level of theory deemed appropriate for our system and the specific questions we aimed to address in this study.

For the electronic transitions of the copper complexes, we agree that these would be of interest, and it is an area we could explore in the future. However, they were not within the scope of this particular study. We appreciate your suggestion, and it will certainly be taken into consideration for our future research.

The used of molecular modeling techniques to elaborate three-dimensional models of the main Zn(II) and Cu(II) complexes on the basis of the experimental evidences has been inserted in the introduction of the revised version of the MS.

I am surprised no paramagnetic shifts were seen in the Ni spectra at pH values where the square planar, diamagnetic complexes are not formed.

Authors reply:

We performed the Ni(II) study as a probe for Cu(II) in a similar coordination mode precisely to characterize the square planar (diamagnetic) complexes. We didn’t probe Ni for the other condition in which it results paramagnetic.

At high pH we have often found that the Cu complex enters a region of slow exchange and so the NMR signals sharpen - ie the spectrum is of the free ligand.

Authors reply:

Unfortunately, we have not observed this phenomenon. For the shorter peptide during titration with successive additions of Cu(II) at high pH, we observed a progressive line broadening. For longer peptides, beside the line broadening, there was the concomitant contribution of their poor solubility which made their analysis more complicated.

The authors calculated the relative binding of Zn and Cu and found Cu is better. This is not really relevant.  What is more important is how stable the Zn/peptide complexes are relative to other endogenous ligands.   

Authors reply:

Thank you for highlighting this important detail. We calculated the relative binding of the studied metals and prepared the competition plot to visualize the situation in equimolar system, however we are aware that such conditions rarely ever happen in vivo. To study the stability of the complexes more deeply, we decided to calculate the dissociation constants, Kd. which refers to the concentration of the free metal ion (expressed in molarity), when half of the ligand exists in a complex form, and another half is not complexed. Dissociation constant is very frequently used to compare the stability of metal complexes with endogenous ligands.  We compared Kd values obtained for our system with those available for some important endogenous ligands- human serum albumin for Cu(II) and Zn(II) and glutathione for Zn(II) (for glutathione we calculated the Kd ourselves using the stability constants available in the literature). This comparison has helped us to understand how the stability of the complexes of our system places among the very efficient biological ligands. The text has been changed accordingly.

The measured equilibrium constants have quite low standard deviations, given the number of titrations performed. Do we have any idea as to the 'goodness of fit' between the experimental and theoretical data.  

Authors reply:

Thank you for this question. Actually, the standard deviations for the obtained stability constants are pretty normal for the potentiometric studies of metal ion-peptide complexes. There are many papers in which the standard deviations are even lower than those that we got from our calculations, for example:

  • Dalton Trans., 2022, 51,14267
  • New J. Chem., 2019, 43, 907-916
  • Dalton Trans., 2014, 43, 16680

Another parameter describing the overall goodness of fit is the sigma parameter [Gans, P.; Sabatini, A.; Vacca, A. Investigation of equilibria in solution. Determination of equilibrium constants with the HYPERQUAD suite of programs. Talanta 1996, 43, 1739-1753, doi:10.1016/0039-9140(96)01958-3.]. In our calculations, sigma values were in the range of 0.8 and 1.8, suggesting a good agreement between experimental and theoretical data. The plot of residuals (differences between the value of an observed quantity and the value of the corresponding calculated quantity) also suggested a good fit of the data, with low residuals present for the start and the finish of the titration, and higher residuals for the middle of the titration, which is a typical behavior for potentiometric studies of such systems.

SPECFIT was used to calculate the electronic spectra, but the equilibrium constants were adjusted as well.  Why? 

Authors reply:

Thank you for asking about this detail. In fact, simultaneous calculation of stability constants and electronic spectra is the standard procedure we use in our lab for the determination of stability constants of Fe(III) complexes for siderophore analogues. In those cases, we do use much more intense and well-defined charge transfer bands, and do not calculate stability constants from potentiometric data. Here, we have used this standard description in the experimental part. We have done this calculation for the Cu(II) complexes with peptide but only to check if the constants calculated from pH-dependent UV-Vis titrations are analogous to the ones calculated by potentiometry, and indeed they were very close. Still, as d-d bands of Cu(II) complexes are rather broad and weak, and the complexes are not very well separated, we decided to fix the stability constants calculated from potentiometric data and calculate only the spectra. This detail was corrected in the experimental section, as shown below.   

“Absorptivities (ε, M−1 cm−1) were calculated at the pH value of maximum concentration of the considered species, as indicated by the potentiometric distribution diagrams. UV−Vis data were refined to obtain the simulated spectra for various Cu(II) complex forms using SPECFIT/32 software that adjusts the absorptivity and the stability constants of the species formed at equilibrium.” was corrected to “To calculate absorptivities (ε, M−1 cm−1) of the various Cu(II) complex forms, as calculated by potentiometry and shown in distribution diagrams, UV−Vis data were refined using SPECFIT/32 software that adjusts the absorptivity and the stability constants of the species formed at equilibrium. As d-d bands of Cu(II) complexes are rather broad and weak, and the complexes are not very well separated, we have fixed the stability constants calculated from potentiometric data and calculated only the spectra.”

Reviewer 2 Report

Dear authors,

Even if I am working on ACE2 enzyme, I am not specialist of its structure and, thus, not highly familiar with the techniques you are presenting. However, I found interesting your approach and your goal. As you mentioned, it is important to find new therapeutical approaches to treat Covid-19. 

Rgarding the manuscript, I have some comments.

1/ In the introduction, I am a little bit disagree with one point (not directly important for your manuscript). Here the two points: 

« In fact, although the massive vaccination campaigns decreased the mortality associated with COVID-19 all over the world and the virus is now circulating freely with symptoms similar to a normal cold or flu,”. I will add the world “significantly” before decreased. Besides, the loss of smell is not something largely found in flu symptoms.

2/ Please separate the results part with the discussion part. It is too complicated to follow as it is now. It migh be better if you separate clearly the results and the discussion to have the possibility to better compared with the litterature.

3/ I did not clearly understand how your results can be used in therapeutics. You mentionned "Whether it is possible to exploit this information in order to arrange a therapeutic strategy could be the topic of a new research and opens new perspectives in the treatment of COVID-19 in those patients who still experience severe symptoms after infection, or in case new aggressive variants would emerge.". I found very correct and honnest to not extrapolate results and to be carefull with it. But, I believe that you can conclude with an hypothesis of how your results can be used as a therapeutic tools.

4/ I noted that the last author (the corresponding one?) is co-authors of 8 of the articles in references. Thus, 15,09% of the cited articles. Following the rules of Clarivates, autocitation for the last author shall not exceed 5%. All the articles you quoted are relevant. But, I believe that you can remove and remplace 1 or 2 of the articles among these 8. 

Otherwise, I congratulate you for this nice work. 

Author Response

Dear authors,

Even if I am working on ACE2 enzyme, I am not specialist of its structure and, thus, not highly familiar with the techniques you are presenting. However, I found interesting your approach and your goal. As you mentioned, it is important to find new therapeutical approaches to treat Covid-19. 

Authors reply:

We thank the reviewer for the positive comments and valuable suggestions. In the revised version, we made any effort to follow the suggestions in order to improve the quality of the MS.

Regarding the manuscript, I have some comments.

1/ In the introduction, I am a little bit disagree with one point (not directly important for your manuscript). Here the two points: 

« In fact, although the massive vaccination campaigns decreased the mortality associated with COVID-19 all over the world and the virus is now circulating freely with symptoms similar to a normal cold or flu,”. I will add the world “significantly” before decreased. Besides, the loss of smell is not something largely found in flu symptoms.

Authors reply:

The adverb “significantly” has been added to the sentence as suggested. The observation that loss of smell is not a common symptom in flu is correct, the very meaning of this sentence was that in this moment Covid-19 has been downgraded to “a normal flu”, meaning that side effects of this viral disease are usually not more severe than those of a seasonal influenza. We have reformulated the sentence, hoping it may be clearer now.

2/ Please separate the results part with the discussion part. It is too complicated to follow as it is now. It migh be better if you separate clearly the results and the discussion to have the possibility to better compared with the literature.

Authors reply:

We do not agree with reviewer 2's suggestion that the results (and therefore all figures and tables) be separated from the discussion of the data. This separation would lead not only to lengthen the manuscript further, but it would in fact be even more complicated for the reader to follow the discussion of the data which would recall from time to time a specific table or a figure already mentioned above. In fact, the structure of these types of articles which are based on a combination of potentiometric and spectroscopic techniques (see for example refs 26, 27, 30-37) is organized exactly as we conceived it (Results and discussion), i.e. gradually showing the results and discussing them point by point with the most appropriate figures to reinforce the interpretation of the data. Furthermore, reviewer 1 also points out that "The paper is well-written and easy to follow". For this reason, we are convinced that the current structure of the article is the most appropriate.

3/ I did not clearly understand how your results can be used in therapeutics. You mentionned "Whether it is possible to exploit this information in order to arrange a therapeutic strategy could be the topic of a new research and opens new perspectives in the treatment of COVID-19 in those patients who still experience severe symptoms after infection, or in case new aggressive variants would emerge.". I found very correct and honnest to not extrapolate results and to be carefull with it. But, I believe that you can conclude with an hypothesis of how your results can be used as a therapeutic tools.

Authors reply:

Since this research work was carried out by a group of chemists and no pharmacologists were included in the research team, we believe we do not have the expertise to suggest a therapy for Covid-19.

Zn plays a crucial role in COVID-19 symptomatology and patients showing zinc deficiencies are the ones experiencing the most severe symptoms. In this study we have highlighted how the interface region of ACE2 for spike protein can be a target for metal ions such as Zn. This binding could affect the binding affinity between the two ACE2/S proteins.

These factors could be connected to each other, but we believe that it would be neither wise nor ethical for us to propose a new therapy without an in-depth study by pharmacologists and physicians. Therefore, in addition to suggesting a supplementation of Zn, especially for people who lack it, we prefer to leave the manuscript as it is without trying to indicate possible therapies which could instead be the subject of a future study involving the most suitable expertise.

4/ I noted that the last author (the corresponding one?) is co-authors of 8 of the articles in references. Thus, 15,09% of the cited articles. Following the rules of Clarivates, autocitation for the last author shall not exceed 5%. All the articles you quoted are relevant. But, I believe that you can remove and remplace 1 or 2 of the articles among these 8. 

Authors reply:

We follow the advice to reduce the autocitations, and in the revised version of the MS two articles have been removed from the bibliography as suggested.

References removed:

  • Zoroddu, M.A.; Kowalik-Jankowska, T.; Kozlowski, H.; Molinari, H.; Salnikow, K.; Broday, L.; Costa, M. Interaction of Ni(II) and Cu(II) with a metal binding sequence of histone H4: AKRHRK, a model of the H4 tail. Biochim Biophys Acta 2000, 1475, 163-168, doi:10.1016/s0304-4165(00)00066-0.
  • Magri, A.; Munzone, A.; Peana, M.; Medici, S.; Zoroddu, M.A.; Hansson, O.; Satriano, C.; Rizzarelli, E.; La Mendola, D. Coordination Environment of Cu(II) Ions Bound to N-Terminal Peptide Fragments of Angiogenin Protein. Int J Mol Sci 2016, 17, doi:10.3390/ijms17081240.

Otherwise, I congratulate you for this nice work.

Authors reply:

We thank you again for the positive feedback.